# A New Approach to Enhanced Swarm Intelligence Applied to Video Target Tracking

**DOI:** 10.3390/s21051903

**Published:** 2021-03-09

**Authors:** Edwards Cerqueira de Castro, Evandro Ottoni Teatini Salles, Patrick Marques Ciarelli

**Affiliations:** Centro Tecnológico, Programa de Pós Graduação em Engenharia Elétrica, Universidade Federal do Espírito Santo, Vitória, Espírito Santo 29075-910, Brazil; evandro.salles@ufes.br (E.O.T.S.); patrick.ciarelli@ufes.br (P.M.C.)

**Keywords:** swarm intelligence, meta-heuristic, dynamic optimization problems, video target tracking, time series forecasts

## Abstract

This work proposes a new approach to improve swarm intelligence algorithms for dynamic optimization problems by promoting a balance between the transfer of knowledge and the diversity of particles. The proposed method was designed to be applied to the problem of video tracking targets in environments with almost constant lighting. This approach also delimits the solution space for a more efficient search. A robust version to outliers of the double exponential smoothing (DES) model is used to predict the target position in the frame delimiting the solution space in a more promising region for target tracking. To assess the quality of the proposed approach, an appropriate tracker for a discrete solution space was implemented using the meta-heuristic Shuffled Frog Leaping Algorithm (SFLA) adapted to dynamic optimization problems, named the Dynamic Shuffled Frog Leaping Algorithm (DSFLA). The DSFLA was compared with other classic and current trackers whose algorithms are based on swarm intelligence. The trackers were compared in terms of the average processing time per frame and the area under curve of the success rate per Pascal metric. For the experiment, we used a random sample of videos obtained from the public Hanyang visual tracker benchmark. The experimental results suggest that the DSFLA has an efficient processing time and higher quality of tracking compared with the other competing trackers analyzed in this work. The success rate of the DSFLA tracker is about 7.2 to 76.6% higher on average when comparing the success rate of its competitors. The average processing time per frame is about at least 10% faster than competing trackers, except one that was about 26% faster than the DSFLA tracker. The results also show that the predictions of the robust DES model are quite accurate.

## 1. Introduction

The goal in optimization problems is the search for a solution that minimizes (or maximizes) a cost function associated with the problem in a set of possible solutions called solution space.

Evolutionary algorithms (EAs) and swarm intelligence algorithms (SIAs) are two important categories of optimization methods. EAs use a population of agents or particles that explore the solution space in search of the optimal solution inspired by the Darwin’s theory of the evolution of species. SIAs use a population of particles that explore the solution space while interacting with each other and with the environment, resulting from this interaction a coherent global pattern [1]. Both algorithms are called meta-heuristics.

The advantages of the swarm intelligence strategy that make it so popular are the simplicity and flexibility of the algorithms, their derivative-free mechanisms, and their avoidance of the local optimum [2].

Some examples of SIAs are: Particle Swarm Optimization (PSO) [3], Shuffled Frog Leaping Algorithm (SFLA) [4], Salps Search Algorithm (SSA) [5], Cuckoo Search (CS) [6], Ant Colony Optimization (ACO) [7], Firefly Algorithm (FA) [8], and Gray Wolf Optimization (GWO) [2]. For a recent up-to-date list of meta-heuristics, see [2].

The common characteristics of SIAs are that they have search engines inspired by nature, are based on populations, and are interactive. Their main difference, apart from the source that inspires their behavior, is the way that the solution space is globally and locally explored by agents [1].

Originally, SIAs were designed for stationary optimization problems in which the optima do not change within the solution space, and the algorithms efficiently converge to the optimum or near-optimum solution. However, in many real-world situations, optimization problems are subject to dynamic environments in which the optimum can change its position within the solution space. A strategy to address this challenge is to adapt SIAs to work in dynamic optimization problems (DOPs) [1] by considering the optimization environments as a sequence of stationary optimization problems.

Video target tracking is the task of estimating the position and trajectory of one or more targets in a digital video image sequence. A digital video is a sequence of ordered images or frames, and a target can be any object of interest in a scene, e.g., one or more people walking on a sidewalk, cars traveling down an avenue, or animals running in the field.

In the last decades, the increasing popularity of video cameras and digital computers, technological advances, and the great extent to which these products are offered on the market have led to an increased interest in automated video analysis for various real-world applications, e.g., surveillance and security [9], human–machine interfaces [10], and robotics [11]. A more detailed introduction to the subject is provided in [12].

Targets can be represented by their shape and appearance. The appearance of a target is characterized by features (e.g., color, texture, corners) that are extracted from a specific region of the target [12].

According to [13], there are two methods of estimating the target’s trajectory, online and batch. The online method uses the current and previous frames to estimate the state of the target in each time period and the batch method uses the entire sequence of frames to optimize the target estimate in each time using past and future information, however, the batch method cannot be used in applications where it is necessary to track the target in real time. In this work we will only deal with online tracking of a single target.

Tracker models can be classified into two categories according to [14]:Category 1 whose models are based on stochastic algorithms. These algorithms are used to predict the position of the target in each frame according to a pattern of movement and observable characteristics of the target. The main examples are the Kalman Filter (KF) [15] and the Particle Filter (PF) [16];Category 2 whose models are based on template matching. These models select regions of the current frame and extract certain observable characteristics from these regions, which are then compared with the respective characteristics of one or more templates of the target to be tracked. The most similar region indicates the likely position of the target in the current frame. A classic tracker in this category is the Mean Shift (MS) [17] and other models in this category can be found in [18]. Therefore, there is a class of trackers whose models are based on optimization algorithms since the likely position of the target in a frame is indicated by the position of maximum similarity between the templates and the candidate target [19,20,21,22,23].

It is worth mentioning that many authors have proposed the use of SIA to optimize the trackers based on PF algorithm [14,24,25]. Furthermore, in recent years, great progress and importance have been given to trackers based on Correlation Filter (CF) [26,27,28] and Deep Learning (DL) [29,30,31] due to the good results in visual tracking.

Video target tracking is a challenging task because of the complexity generated by interference from several factors [32], such as: partial or total occlusion of the target; sudden changes in ambient lighting and movement of the target; rotations, deformations and scale changes of the target; blurred and noisy digital images; videos recorded by low-resolution cameras; and image backgrounds with similar aspects to the target. The interference of factors modifies the optimization environment. Therefore, the video target tracking problem is a particular case of a DOP.

The objective of this work is to propose, analyze and discuss a DOP-enhanced SIA approach applied to video target tracking. A tracker based on the meta-heuristic Shuffled Frog Leaping Algorithm (SFLA) [4] is proposed and adapted to DOPs, the Dynamic SFLA tracker (DSFLA).

The proposed method introduces a procedure to select some good solutions from the previous frame and consider them in the next frame, maintaining the diversity of solutions and an adaptive transfer procedure for the selected solutions. Another procedure of the proposed method is the delimitation of the solution space in a promising region of the image by forecasting time series. A version of the double exponential smoothing (DES) [33] model that is robust to outliers is used to forecast the position of the target and delimit the solution space in a promising region for a more efficient search.

The quality of the tracking will be measured by the area under curve (AUC) of the success rate per Pascal metric or One-Pass Evaluation (OPE). The average processing time per frame will also be analyzed.

The quality results of the DSFLA tracker will be compared with the quality results obtained from other SIA-based video target trackers: the PSO tracker whose meta-heuristic is Bare Bones PSO (BBPSO) [34], the Adaptive Discrete Swarm Optimization (ADSO) tracker [22], the SFLA tracker whose meta-heuristic is SFLA [4] and the SSA tracker [23] whose meta-heuristic is SSA [5].

The innovations and contributions presented in this work are:A new swarm intelligence algorithm for dynamic optimization problems;A new video target tracker;An appropriate algorithm for optimization problems with discrete solution space;A new and adaptive method of knowledge transfer between two optimization environments and the reduction of the solution spaces based on an efficient time series forecast model;A case study in the area of video target tracking showing results compatible with state-of-the-art models based on SIA;In situations of controlled ambient light, small occlusions and little camouflage of the target, the DSFLA is fast and stable in tracking any target. Especially, it is robust in situations where there are rotations or fast movements of the target, in low resolution videos corrupted by blurred or noisy interference.

Table 1 presents the summary of characteristics and the strengths and weaknesses of the trackers treated in this work. For all trackers covered in this work, the common characteristics are: they were designed for online tracking of a single target and are based on SIA for optimization problems. They are general propose trackers.

The next section presents a summary of the latest works related to the proposed theme.

## 2. Related Work

Canalis et al. [19] were one of the first to apply the PSO algorithm to a video target tracking issue. The results were promising, comparable to the results of the traditional Mean Shift (MS) [17] and Particle Filter (PF) Bootstrap [16] trackers. In [20], an improved version of the PSO produced results that outperformed those in [19]. However, both approaches used the meta-heuristic in a stationary optimization environment and there was no delimitation of the solution space in a region of the image.

Gao et al. [21] proposed a tracker based on the Cuckoo Search (CS) [6] algorithm. The CS algorithm mimics the predatory behavior of the cuckoo bird in relation to the laying of eggs during the nesting period. They used six challenging videos and the Bhattacharyya distance [17] from the color histogram based on a space kernel as a measure of similarity between targets. They compared the performance of the CS tracker with the PF, MS, PSO and more four versions of the PSO trackers, and the CS outperformed the other competitors in terms of processing time and tracking quality using the Euclidean distance from the central points of the estimated and true targets. However, similarly to the previously mentioned trackers [19,20], the CS tracker considers only stationary optimization environments.

Bae et al. [22] presented the Adaptive Discrete Swarm Optimization (ADSO) algorithm, a tracker that applies solution space delimitation in a version of the PSO algorithm for the discrete space of solutions. PSO was originally designed for stationary optimization problems with continuous solution space, therefore, when it is applied to discrete optimization problems, the chance of the PSO converging prematurely to local optimum is greater [22]. ADSO was the first video target tracking algorithm that employed the swarm intelligence method in DOPs for discrete solution spaces.

ADSO works in dynamic optimization environments by transferring knowledge from one frame to the other through a probability function that controls the diversity of the particles. This function assigns a probability to each particle according to the degree of occlusion of the target, as defined by two thresholds, $th_{1}$ and $th_{2}$ where $0\leq th_{1}<th_{2}\leq 1$. These probabilities define whether each particle variable will receive the value of the optimum or if it will receive a random value that covers the entire amplitude of the variable within the solution space.

Bae et al. [22] used seven videos from the public benchmark Pami (http://sites.google.com/site/benchmarkpami/, accessed on 24 November 2020) and the Bhattacharyya distance from the HSV color histogram as a measure of similarity between targets. The results of ADSO outperformed those of PSO trackers and another EA-based tracker in terms of processing time and Euclidean distance x and y coordinates between estimated and true targets. The results showed that the ADSO is good for tracking fast-moving targets.

Zang et al. [23] presented a tracker based on the Salps Search Algorithm (SSA) [5]. The SSA meta-heuristic mimics the behavior of a group of salps swimming and foraging in the deep ocean. The salp is a member of the Salpidaes family; it has a transparent barrel-shaped body and swims by propulsion, forming long chains. This chain (particles) is formed by a leader who seeks a source of food (the optimal solution) and by followers who follow the movement of the leader. The movement of the leader is responsible for the global exploration of the solution space while the movement of the followers is responsible for the local exploration. The algorithm performs L search iterations in the solution space and, at each iteration, the movement of the leader is controlled by a function that makes a balance between local and global exploration.

Zang et al. [23] used 13 videos from the public benchmark (in http://www.visual-tracking.net, accessed on 24 November 2020) and the cross-correlation coefficient of the Histogram Oriented Gradient (HOG) characteristic [35] as a measure of similarity between targets.

The results in [23] outperformed another ten state-of-the-art trackers in terms of performance quality and speed. However, the SSA tracker is based on optimization in stationary environments and does not delimit the solution space in a region of the image.

## 3. Architecture

### 3.1. Swarm Intelligence Algorithm in a Dynamic Optimization Problem

Optimization problems entail searching for an optimal (or near-to-optimal) solution among a set of feasible solutions. This search may or may not be subject to one or more restrictions.

Solutions are made up of variables associated with the problem. A solution is feasible if the values assumed by the variables satisfy the restrictions. A feasible solution is optimal if it minimizes or maximizes the objective function (or fitness function or cost function), which measures the quality of solutions.

SIAs reproduce the collective intelligence that emerges from the behavior of a group of agents and are inspired by nature. SIAs were designed for stationary optimization problems in which the parameters, the solution space, and the objective function do not change during the optimization process. However, in many real-world situations, optimization problems are subject to dynamic environments in which the optimum can change its position within the solution space during the optimization process. The optimization environment of a DOP is more challenging than that of a stationary optimization problem since repeated optimization is required in the presence of a changing optimum [1].

A DOP can be defined as a sequence of stationary problems that need to be optimized and can be formally described as follows: Optimize f(p,t) subject to
(1)P(t)⊆S,t∈T,
where S is the solution space, t is the time, and
(2)f:T×S→ℝ
is the objective function that associates a real number to each solution p∈S. P(t) is the set of feasible solutions over time t. Each feasible solution p∈P(t)⊆S consists of a vector of d dimensions $p=\left(p_{1},p_{2},\ldots ,p_{d}\right)$, where each component of this vector corresponds to a variable of the problem.

Each feasible solution in P(t) is associated with a set of neighbors N(p)⊆P(t) and the feasible solution $p^{\prime }\in N\left(p\right)$ is a local optimum if and only if
(3)$$f\left(p^{\prime },t\right)\leq f\left(p,t\right),\forall p\in N\left(p\right),$$
is a minimization or
(4)$$f\left(p^{\prime },t\right)\geq f\left(p,t\right),\forall p\in N\left(p\right),$$
if it is a maximization.

Similarly, the feasible solution $p^{*}\in N\left(p\right)$ is a global optimum if and only if
(5)$$f\left(p^{*},t\right)\leq f\left(p,t\right),\forall p\in P\left(t\right),$$
is a minimization or
(6)$$f\left(p^{*},t\right)\geq f\left(p,t\right),\forall p\in P\left(t\right),$$
if it is a maximization.

The drawback of the SIAs is that the convergence ability decreases the particle diversity, reducing the ability of the algorithm to adapt to a new optimization scenario. On the other hand, for an SIA to adapt to a DOP, it is necessary to promote the transfer of knowledge. However, if too much knowledge is transferred, then the optimization process in the current environment may begin near a poor location and get trapped in a local optimum [1].

The goal is to promote an ideal balance between knowledge transfer and the diversity of particles since they constitute two conflicting factors [1]. Therefore, enhanced SIAs that promote this balance are suitable for dynamic optimization.

There are a few ways to promote this enhancement, e.g., maintaining a memory scheme of the best particles from previous optimizations and using them in the current optimization or maintaining multiple populations and allocating them to different regions of the solution space [1].

### 3.2. The Meta-Heuristics SFLA and BBPSO

More details of the SFLA and BBPSO algorithms will be given in this subsection since three of the five trackers were based on them, whereas the ADSO and SSA trackers were reproduced in this work.

#### 3.2.1. The Shuffled Frog Leaping Algorithm

The memetic meta-heuristic SFLA was proposed by Eusuff et al. [4] to solve combinatorial optimization problems. Its solution space exploration mechanism mimics the behavior of a group of frogs (the particles) in a swamp (the solution space) as they vie for the best place to feed. The best places are stones (solutions to the problem), which are located at discrete points of the swamp.

The SFLA starts by randomly generating virtual frogs in the swamp and grouping them into frog communities called memeplexes. Frogs jump within the solution space and are influenced by the positions of the frog with the best fitness in each memeplex and the frog with the best fitness in the swamp.

The position of the worst frog in each memeplex is changed according to
(7)$$p_{w}^{\tau }=p_{w}^{\tau -1}+l_{w}^{\tau },$$
where $p_{w}^{\tau -1}$ is the position of the worst memeplex frog in the previous iteration, $p_{w}^{\tau }$ is the position of the worst memeplex frog and $l_{w}^{\tau }$ is the jump made by the worst frog in the memeplex in iteration τ ($\tau =2,3,\ldots ,n_{\tau }$, where $n_{\tau }$ is the maximum number of iterations). The jump is limited by a constant positive predefined and problem-dependent threshold $l_{Max}$ ($-l_{Max}\leq l_{w}^{\tau }\leq l_{Max},\forall \tau$).

To calculate the worst frog jump, the SFLA calculates a new position adding a random jump towards the best frog in the memeplex as follows
(8)$$l_{w}^{\tau }=U\left(p_{b}^{\tau -1}-p_{w}^{\tau -1}\right),$$
where $p_{b}^{\tau -1}$ is the position of the best memeplex frog in the previous iteration and U is a pseudo-random number uniformly distributed over a continuous unit interval.

If the fitness of the new position of the worst frog is not improved, the jump calculated by Equation (8) is discarded and another random jump is added to the original position of the worst frog in the memeplex towards the best frog in the swamp as follows
(9)$$l_{w}^{\tau }=U\left(p_{g}^{\tau -1}-p_{w}^{\tau -1}\right),$$
where $p_{g}^{\tau -1}$ is the position of the best swamp frog in the previous iteration.

If the new jump does not improve the fitness of the worst frog in the memeplex, then it is replaced by a new frog located at a random point in the swamp.

After performing the jumps and updating the fitness of each frog, they are randomly redistributed among memeplexes before the next iteration of the algorithm. The procedure is repeated until the stop condition is reached.

The algorithm performs simultaneously an independent local search in each memeplex. The global search is guaranteed by the shuffling of frogs and the reorganization of memeplexes. The algorithm also generates random virtual frogs to increase the opportunity for new information in the population [4].

The main advantages of SFLA are that it is more powerful in solving complex combinatorial optimization problems, has a faster search capability, and is more robust in determining the global optimum because of the evolution of several memeplexes (the structure responsible for local exploration) and the scrambling process (the structure responsible for global exploration), which can improve the quality of individuals [36]. The pseudocode and more details are provided in [4].

#### 3.2.2. The Bare Bone Particle Swarm Optimization

The Bare Bone PSO (BBPSO or Gaussian PSO) [34] meta-heuristic, as in the classic PSO, mimics the behavior of a flock of birds (particles) that fly over the solution space while exchanging information with their neighbors, and it has the advantage of working with only two parameters: the number of particles and the neighborhood topology [37].

There are two types of neighborhood topology, the global one in which the particles communicate with each other, and the local where each particle communicates with a subgroup of particles. In this work, we adopted the local neighborhood topology.

The main difference between the two versions of the PSO is that BBPSO uses a Gaussian random variable to update the position of the particles instead of adding a velocity equation, as occurs in the classic PSO.

The equation for updating the position of the particles in the BBPSO is given by
(10)$$p_{i}^{\tau }=\mu_{i}^{\tau }+\sigma_{i}^{\tau }⊗Z$$
with
(11)$$\mu_{i}^{\tau }=\frac{pBest_{i}^{\tau -1}+gBest^{\tau -1}}{2},$$
(12)$$\sigma_{i}^{\tau }=\left|pBest_{i}^{\tau -1}-gBest^{\tau -1}\right|,$$
where ⊗ is the element-by-element product between two vectors, $p_{i}^{\tau }$ is the particle i in iteration τ with dimension d; $\mu_{i}^{\tau }$ corresponds to the mean and $\sigma_{i}^{\tau }$ the variance of the random vector Z with Gaussian distribution. $pBest_{i}^{\tau -1}$ is the best position visited by the particle i until the iteration τ−1 end $gBest_{i}^{\tau -1}$ is the best position visited by the swarm until the iteration τ−1 ($\tau =1,2,\ldots ,n_{\tau }$ and i=1,2,…,n, with $n_{\tau }$ the number of iteration and n the number of particle).

The pseudocode and more details are provided in [34].

### 3.3. Target Tracking

Target tracking is a particular case of a DOP since the challenges in the scene modify the solution space, and the optimal solution can vary in each frame.

In this work, to ensure a fair experiment for all trackers, the particles of the meta-heuristics are represented by rectangular bounding boxes and are denoted by four-dimensions vectors p=(x,y,w,h), where (x,y) is the 2D coordinate of the pixel located in the upper left corner of the bounding box, and (w,h) denotes the horizontal and vertical dimensions, referring to the base and height of the bounding box, respectively. Each particle corresponds to a candidate target.

The appearance and characteristics of the targets are represented by the standardized histogram of the first channel of the YCbCr color model [12]. From previous experience, the inclusion of the second and third YCbCr channel histograms result in an almost zero gain of target discrimination power at the expense of a higher computational cost due to a longer processing time. Therefore, we decided to work only with the first channel for a lower computational cost.

A standardized histogram is a unit area histogram and is an asymptotically unbiased and consistent estimator of the probability density function [38]. The choice of the standardized color histogram is due to its invariance to rotations and scale changes [39], in addition to being a quick approach.

To measure the similarity between the candidate targets and the template, the Bhattacharyya distance [17] is adopted. The equation is given by
(13)$$\beta \left(H_{T}^{*}\left(b\right),H_{P}^{*}\left(b\right)\right)=\sqrt{1-\sum_{b=1}^{nBins}\sqrt{H_{T}^{*}\left(b\right)H_{P}^{*}\left(b\right)}},$$
where $H_{T}^{*}\left(b\right)$ and $H_{P}^{*}\left(b\right)$ are the standardized histograms of the template and the candidate target, respectively, b indicates the histogram bin, and *nBins* indicates the total number of bins in the histogram.

The Bhattacharyya distance is a standardized measure that is limited to the continuous unit interval, where zero indicates total similarity and 1 indicates the total lack of similarity between histograms.

Video target trackers in category 2 of the classification given in [14] based on SIAs work as follows: In each frame, n particles are scattered at random within the solution space given by
(14)S={∀p∈ℤ+4|1≤x,w≤R,1≤y,h≤Q},
where R and Q are the total pixels of the horizontal and vertical dimensions of the frame, respectively.

Then, the meta-heuristic moves the particles and updates their fitness until a stop condition is reached (e.g., a maximum number of iterations or a minimum quality value of the best solution). When the meta-heuristic reaches the stop condition, the algorithm returns the best quality particle of the swarm, $gBest=(x_{g},y_{g},w_{g},h_{g})$ where gBest is $p^{*}$ or $p^{\prime }$, indicating the target’s position in the current frame. However, it is possible to take advantage of the good solutions of the previous frame to set the initial location of particles in the current frame. It is also possible to spread the particles over a limited region of the solution space since the hypothesis that the target does not move a long distance from one frame to another one is plausible in the vast majority of cases of target tracking in videos.

### 3.4. Robust Double Exponential Smoothing

One of the proposals of this work is to delimit all dimensions of the solution space. The goal is to surround the target in the next frame in a promising region and increase the chances of detection. A robust version of the DES [33] time series model was used for this purpose.

The exponential smoothing model, also called the Holt and Winters model [33], works on a time series by decomposing it into four factors: level, trend, seasonal factor and an unpredictable residual factor called random noise.

The process of estimating these factors is based on exponential smoothing, i.e., the process eliminates sudden variations in the observed series, and it is then described by its structural components (the four factors). The factor estimation process involves the calculation of weighted arithmetic averages in which the weights decay exponentially over time as it moves to the past values of the time series. More details about the theoretical issues involving Holt and Winters modeling are in [40], and for a review on the subject, see [41,42].

The double exponential smoothing (DES) model decomposes a stochastic process ${\left\{Z_{t}\right\}}_{t\in T}$ into the level, the trend and a random error term according to
(15)$$Z_{t}=\mu_{t}+M_{t}+\varepsilon_{t},\forall t\in T,$$
where $Z_{t}$ is the random variable of the stochastic process ${\left\{Z_{t}\right\}}_{t\in T}$ at time t defined in the same sample space, $\mu_{t}$ is the smoothing factor that corresponds to the level at time t, $M_{t}$ is the smoothing factor that corresponds to the trend over time t, and $\varepsilon_{t}$ is a random variable with a zero mean and constant positive variance and is not correlated with $\varepsilon_{\tau },\;\forall \tau \neq t$ and $Z_{t},\;\forall t$. 

Estimates of level, denoted by ${\hat{\mu }}_{t}$, and the trend, denoted by ${\hat{M}}_{t}$, are given by, respectively,
(16)$${\hat{\mu }}_{t}=\alpha_{3}z_{t}+\left(1-\alpha_{3}\right)\left[{\hat{\mu }}_{t-1}+{\hat{M}}_{t-1}\right],$$
(17)$${\hat{M}}_{t}=\alpha_{4}\left({\hat{\mu }}_{t}-{\hat{\mu }}_{t-1}\right)+\left(1-\alpha_{4}\right){\hat{M}}_{t-1},$$
where the coefficients $\alpha_{3}$ and $\alpha_{4}$ are called smoothing constants ( $0<\alpha_{3}<1;\;0<\alpha_{4}<1$), and the higher the value of the coefficients, the lower the weight that is given to the past values of each factor; $z_{t}$ is the current value of the observed series; ${\hat{\mu }}_{t}$ is the current time-smoothing value used to estimate the level; and ${\hat{M}}_{t}$ is the current trend estimate. When t=1, it is necessary to set the starting values of ${\hat{\mu }}_{1}$ and ${\hat{M}}_{1}$ (in general, but not necessarily) to ${\hat{\mu }}_{1}=z_{1}$ and ${\hat{M}}_{1}=0$.

The time horizon forecasts k from the instant t are given by
(18)$${\hat{Z}}_{t}\left(k\right)={\hat{\mu }}_{t}+k{\hat{M}}_{t},$$
where ${\hat{Z}}_{t}\left(k\right)$ is the forecast value of the random variable $Z_{t+k}$ of the generating process of the observed series ${\left\{Z_{t}\right\}}_{t\in T}$.

The impact of an outlier on the series forecast can be seen by observing (16) and (17). When an outlier $z_{t}$ is observed, the values of ${\hat{\mu }}_{t}$ and ${\hat{M}}_{t}$ are overestimated. However, these values continue to affect future estimates at both the level and the trend, producing persistently skewed forecasts.

In order to mitigate the effects of outliers on predictions, we used a version of the DES model that is robust to outliers. In this case, the observation of the series at time t, $z_{t}$, is replaced by the lower limit value, $LL_{t}$, or the upper limit value, $UL_{t}$, when $z_{t}<LL_{t}$ or $z_{t}>UL_{t}$, respectively. Limit values are calculated and updated at every time t according to, respectively,
(19)$$LL_{t}={\bar{Z}}_{t}^{*}-3\sqrt{s_{t}^{*}},$$
(20)$$UL_{t}={\bar{Z}}_{t}^{*}+3\sqrt{s_{t}^{*}},$$
where ${\bar{Z}}_{t}^{*}$ is the average of the observed series, from its update after the first observation until $z_{t}$, and calculated according to
(21)$${\bar{Z}}_{t}^{*}=\frac{t-1}{t}{\bar{Z}}_{t-1}^{*}+\frac{1}{t}z_{t},\;t\geq 1,$$
and $s_{t}^{*}$ is the variance of the observed series, from its update after the first observation to the observation $z_{t}$, and calculated according to
(22)$$s_{t}^{*}=\frac{t-1}{t}\left(s_{t-1}^{*}+{\left({\bar{Z}}_{t-1}^{*}\right)}^{2}\right)+\frac{1}{t}{\left(z_{t}\right)}^{2}-{\left({\bar{Z}}_{t}^{*}\right)}^{2},\;t\geq 1,$$
for t=1, ${\bar{Z}}_{0}^{*}=0$ and $s_{0}^{*}=0$.

Therefore, the robust DES model (RDES) is the model given by Equation (15), the forecast is given by Equation (18), and the estimates of $\mu_{t}$ and $M_{t}$ are given by Equations (16) and (17), respectively. In fact, the RDES model differs from the DES model only when an outlier is observed and the value $z_{t}$ is replaced by the limit values given by Equation (19) or Equation (20).

To measure the quality of forecasts in time series models, it is common to adopt the square root of the mean squared error of the forecast (RMSE) [43] as a metric. The RMSE is calculated for each point coordinate (x,y,w,h) of gBest according to
(23)$$RMSE=\sqrt{\frac{1}{n_{t}}\sum_{t=1}^{n_{t}}{\left(gBest_{t}-{\hat{p}}_{t-1}\left(1\right)\right)}^{2}},$$
where $n_{t}$ is the maximum number of times (or frames) and ${\hat{p}}_{t-1}\left(1\right)$ is the gBest forecast for time t from time t−1 (${\hat{p}}_{t-1}\left(1\right)$ correspond to ${\hat{Z}}_{t-1}\left(1\right)$ of Equation (14)).

The mean squared error measures the variance and squared bias of the forecast for each coordinate [43]. The lower the RMSE, the more homogeneous and less biased the forecast is.

The Euclidian distance of the 2D coordinates (x,y) between the points in the upper left corner of the gBest and ${\hat{p}}_{t-1}\left(1\right)$ bounding boxes will also be measured to check the quality of the forecasts. The shorter the distance, the more accurate the forecast.

## 4. The Proposed Method

The video target tracking model proposed in this work, the Dynamic Shuffled Frog Leaping Algorithm tracker, belongs to category 2 of the classification given in [14]. The DSFLA tracker is an enhanced version of the SFLA meta-heuristic [4] for DOPs. The method also involves a scheme for delimiting the space of solution according to the position of the estimated and predicted targets in the previous frames. To predict the target’s position, the RDES [33] is used.

The DSFLA tracker performs the following steps: for the first frame, n particles are randomly scattered in the solution space S given by Equation (14); for the other frames, the particles are randomly scattered in the delimited solution space by RDES.

Then, the value of the objective function is calculated for all particles, and the SFLA meta-heuristic is executed until a stop condition is reached. In this work, the stop condition is either when the fitness of the best-fit particle of the swarm is less than 0.005 (Bhattacharyya distance) or when the maximum number of iterations is performed.

The solution space (14) is delimited by proposed DSFLA tracker considering the union of two rectangular regions, one around the target estimated by $gBest=\left(x_{g},y_{g},w_{g},h_{g}\right)$ (a region generated by simulating a random walking movement) and another around the predicted target $\hat{p}=\left(\hat{x},\hat{y},\hat{w},\hat{h}\right)$ ($\hat{p}$ correspond to ${\hat{p}}_{t-1}\left(1\right)$ via RDES). The delimitation is given as follows:
The limits of the coordinates (x,y) of the solution space are given by
(24)$$Min\left(x_{g}-\alpha_{1}w_{g},\hat{x}-\alpha_{1}\hat{w}\right)\leq x\leq Max\left(x_{g}+\alpha_{1}w_{g},\hat{x}+\alpha_{1}\hat{w}\right),$$
(25)$$Min\left(y_{g}-\alpha_{2}h_{g},\hat{y}-\alpha_{2}\hat{h}\right)\leq y\leq Max\left(y_{g}+\alpha_{2}h_{g},\hat{y}+\alpha_{2}\hat{h}\right),$$
where $0<\alpha_{1}\leq 2$ and $0<\alpha_{2}\leq 2$ are predefined constants (problem dependent);The limits of the coordinates (w,h) of solution space are given by
(26)$$\gamma_{1}w_{g}\leq w\leq \gamma_{2}w_{g},$$
(27)$$\gamma_{1}h_{g}\leq h\leq \gamma_{2}h_{g},$$
where $0<\gamma_{1}\leq 2$ and $0<\gamma_{2}\leq 2$ are predefined constants (problem dependent).

Figure 1 shows an example of delimiting the solution space obtained from the tenth frame of the video BlurBody (this video was selected from the public Hanyang visual tracker benchmark [44]). The blue bounding box corresponds to the ground truth, and the magenta bounding box corresponds to the estimated target, $gBest=\left(x_{g},y_{g},w_{g},h_{g}\right)$. The green-filled region corresponds to the delimitation of S around the gBest centered in the upper left corner of the blue bounding box, and the yellow-filled region corresponds to the delimitation of S around the $\hat{p}$ centered on the $\left(\hat{x},\hat{y}\right)$ coordinates. The area delimited by the red rectangular box corresponds to the delimited S region in Equations (24) and (25) of the proposed method for (x,y) coordinates.

The new particle selection process selects the particles so that they are distant from each other by a minimum distance, $\delta_{Min}$, calculated as follows:(28)$$\delta_{Min}=\frac{Max\left(S_{w}^{\prime },S_{h}^{\prime }\right)}{\alpha_{0}n},$$
where $S_{w}^{\prime }$ and $S_{w}^{\prime }$ are the vertical and horizontal dimensions of the delimited solution space obtained by Equations (24) and (25), respectively, $\alpha_{0}$ is a predefined maximum percentage of transfer, and n is the number of particles. 

The choice of particles is made in ascending fitness order starting with gBest.

The new adaptive particle transfer process generates n random particles in the bounded solution space and sorts the particles in decreasing order of fitness. Then, the fitness of the selected particles is recalculated and compared with that of the worst generated particles, and the one with the worst fitness is discarded.

The particle transfer process is adaptive since not all $\alpha_{0}n$ particles are selected and that not all selected particles are used (transferred). It varies from frame to frame. In addition, it is possible that some of the transferred and selected particles are positioned outside the delimited solution space but within the solution space S (Equation (10)). For a summary of the proposed DSFLA algorithm, see the pseudocode in Algorithm 1.

**Algorithm 1.** DSFLA tracker’s pseudocode
1: for f = 1:N % for all N frames2:      Img = readframe(f);3:      for i = 1:n % for all n particles4:            if f == 15:                  Generate n particles in S using Equation (14);6:            else7:                  Generate n particles in reduced S;8:            end9:            Extract the histogram and update the fitness;10:      end11:      if f > 112:            Knowledge transfer process;13:      end14:      Update gBest;15:      while <stop condition == false> do16:            Execute SFLA meta-heuristic;17:      end18:      Record gBest and tracker’s performance measures;19:      Particle selection process;20:      Reduce S as a function of gBest: S1;21:      Calculate the forecast pHat via RDES;22:      Reduce S as a function of pHat: S2;23:      Calculate reduced S as a function of S1 and S2 using Equations (24) and (25);24: end

## 5. Experiments and Results

### 5.1. Experimental Design

To investigate the efficiency of the proposed approach, a random sample of 15 videos was selected from the public Hanyang visual tracker benchmark [44] with the respective ground truth with hand-marked targets. The benchmark focus is on tracking a single target online. The benchmark presents 100 videos with generic scenarios and annotations of ground truth for all frames and annotations of attributes that affect the performance of the tracker in identifying the targets.

The public Hanyang benchmark was designed with a collection of video sequences most commonly used in object tracking. It contains videos from various datasets such as the VIVID [45], CAVIAR (http://homepages.inf.ed.ac.uk/rbf/CAVIARDATA1, accessed on 24 November 2020). Other benchmarks such as PAMI share some public videos in common with the Hanyang benchmark.

Table 2 shows the selected videos with the following information: the video’s size (in number of frames), the resolution of the image (in number of pixels) and the main challenges present in the scene. The challenges are rotation in the image plane (IPR), rotation outside the image plane (OPR), fast movement (FM), blurred movement (BM), low resolution (LR), scale variation (SV), deformation of the target (DEF), confusion between the target and background of the image (BC) and occlusion (OCC).

The Hanyang benchmark also includes most of the publicly available codes. The benchmark disseminates performance metrics for in-depth analysis of tracking algorithms. The metrics proposed in [44] are the AUC of the success rate per Pascal metric (success rate) and the Euclidean distance from the central points of the bounding boxes (accuracy).

The Pascal metric [46] is defined according to
(29)$$Pa\left(\xi_{GT},\xi_{C}\right)=\frac{\left|\xi_{GT}\cap \xi_{C}\right|}{\left|\xi_{GT}\cup \xi_{C}\right|},$$
where $\xi_{GT}$ is the bounding box that corresponds to the ground truth, and $\xi_{C}$ is the bounding box that corresponds to the candidate target.

The Pascal metric measures the quality of tracking by quantifying the percentage of pixels that are shared between the bounding boxes, i.e., the overlap of the targets. The Pascal metric ranges from 0.0, when there is no overlap between bounding boxes, to 1.0, when there is total overlap between targets. A target is considered to be detected when the Pascal measure of the candidate target is equal to or greater than a predetermined threshold (in this work the Paschal threshold is 0.5).

The success rate per Pascal metric is the curve formed by the percentages of frames in which the target was detected in a given video, with the threshold of the Pascal metric varying from 0.0 to 1.0. The advantage of observing the curve is that the tracker’s performance is visualized for all thresholds of the Pascal measurement. Therefore, calculating the AUC of the success rate per Pascal metric is a more robust and complete measure to assess the quality of the tracker’s performance when compared with a value for a single fixed threshold. The AUC ranges from 0.0 to 1.0, and the closer it is to 1.0, the better the tracker’s performance. More details on the Pascal metric and the success rate per Pascal metric are provided in [44,46].

There are three tests to assess the robustness of the trackers in [44], the OPE (One-Pass Evaluation) which tests the tracker for the success rate and accuracy from the first to the last frame of the video and the template being the ground truth of the first frame; the TRE (Temporal Robust Evaluation) that tests the tracker using a sequence of frames starting from any frame until the last one; and the SRE (Spatial Robust Evaluation) in which the template is modified from 0.8 to 1.2 of its original scale and starting from 12 different locations in the first frame.

In this work, the performance of the trackers will be evaluated by OPE robustness of the AUC of the success rate per Pascal metric and the average processing time per frame.

The overall results will be summarized by the mean, median and coefficient of variation. The coefficient of variation (cv) is the ratio between the sample standard deviation, s, and the sample mean, $\bar{p}$, of an observed variable. The cv is a dimensionless measure of dispersion and can be expressed as a percentage of variation.
(30)$$cv=\frac{s}{\bar{p}}.$$

### 5.2. Configuration of the Tracker’s Parameters

The configuration of the parameters of each tracker was based on previous experience. The average processing time and the AUC of the success rate per Pascal metric were analyzed to determine the configuration of the parameters of each tracker that results in the best performance.

For this purpose, an analysis was conducted by performing the following experiment: (i) four videos were selected at random from [44]; (ii) the values of the processing time and AUC variables were calculated by averaging three executions of each video for each tracker; (iii) the configuration of the parameters for each tracker was chosen according to the best values of the two metrics.

The videos chosen in this stage of the experiment were Couple and Deer (videos 7 and 9 in Table 1, respectively), Bolt2 (frames: 293; resolution: 480 × 270; challenges: IPR, DEF, BC) and Football1 (frames: 74; resolution: 352 × 288; challenges: IPR, OPR, BC).

It is worth mentioning that the chosen values of the parameters were kept fixed throughout the experiment. The final configuration of parameters for each tracker was as follows:PSO: 150 particles and 15 local groups with 10 particles;ADSO: 40 particles, the thresholds $th_{1}=0.01$, $th_{2}=0.45$, $th_{3}=0.005$, and the probability $H_{f}=0.7$;SFLA: 50 particles, 10 memeplexes with 5 particles, a maximum number of iterations of 10 and a maximum pixel number for frog leaping of 10;DSFLA: The same parameters of SFLA plus $\alpha_{0}=0.3$, $\alpha_{1}=\alpha_{2}=1$, $\alpha_{3}=\left[0.9,0.9,0.8,0.8\right]$, $\alpha_{4}=\left[0.4,0.4,0.5,0.5\right]$, $\gamma_{1}=0.9$, and $\gamma_{2}=1.1$;SSA: 80 particles, a maximum number of iterations of 100, and 20 leader salps.

### 5.3. Analysis of Results

The results of the main experiment are summarized in Table 3 and Table 4. The values presented in the tables correspond to the average of six executions of each video for each of the trackers: PSO, ADSO, SFLA, DSFLA and SSA, always in that order. Replications of all the videos for each tracker were coded in MatLab and executed on the same processor (Intel Pentium Dual-Core, 1.86 GHz, 2 GB DDR2 and 160 GB HDD) to compare the average processing time per frame (it is worth saying that the program codes are not optimized). The average, median, and cv of the 15 videos are in the last three rows of the tables.

Table 3 shows the performance of the trackers according to the average processing time per frame of each video.

As observed in Table 3, the SSA tracker takes more time for execution, and the ADSO tracker is the fastest, however, the DSFLA tracker is, on average, the second fastest. The values are representative given that the cv of the trackers is low, except for the PSO tracker.

Figure 2 shows the Whiskers boxplot (output of MatLab’s internal boxplot function) of each tracker for the observed data of the average processing time per frame. Each boxplot segment corresponds to 25% of the observed values, and the small circles correspond to the outliers. The darker part of the central region of the boxplot represents the interquartile range (IQR), $q_{3}-q_{1}$, where $q_{3}$ is the third quartile and $q_{1}$ is the first quartile. The central point of this region corresponds to the median of the observed values, and the triangles represent the extremes of the 95% confidence interval centered on the median [47], which is calculated according to
(31)$$q_{2}\pm \frac{1.57\left(q_{3}-q_{1}\right)}{\sqrt{n}},$$
where $q_{2}$ is the second quartile, i.e., the median, and n is the size of the observed sample.

If the intervals do not overlap, then we can conclude with 95% confidence that there is a significant difference between the medians, this is equivalent to a statistical test in which the hypothesis that there is no difference between the medians is rejected at 5% significance.

From the graph in Figure 2, all processing times are significantly different except for the those between the SFLA and DSFLA trackers. However, empirically, the proposed DSFLA tracker is systematically about 10% faster than the SFLA tracker.

Table 4 shows the performance of the trackers in relation to the tracking quality according to the AUC of the success rates per Pascal metric. We can see from Table 4 that the cv of all the trackers indicates a low variation of the results except for the PSO tracker. Therefore, we can say that the trackers are satisfactorily stable. Table 4 also shows that the videos 2, 8, 10 and 15 presented the most difficult challenges for all trackers.

Figure 3 shows the boxplots for the AUC of the success rates per Pascal metric. The DSFLA tracker is significantly superior to the PSO and ADSO trackers since the 95% confidence interval referring to the DSFLA tracker has no overlap with the confidence intervals referring to PSO and ADSO trackers. It is not possible to reject the hypothesis of comparable quality between the DSFLA, SFLA, and SSA trackers. However, Table 4 shows empirically that DSFLA tracker results are consistently better than SFLA and SSA trackers results, at about 7.2% higher AUC on average.

Figure 4 and Figure 5 show two examples, chosen at random, of tracking performance given by the success rate per Pascal metric for all trackers for videos 4 and 7, respectively. In Figure 4, the curve that represents the performance of the DSFLA tracker is largely above the other curves. This indicates that DSFLA has higher target detection rates for most of the Pascal metric threshold.

The graphs of the success rate per Pascal metric for most of the other videos show results that reflect the performance of the trackers shown in Figure 5.

The DSFLA tracker produced the best results with videos 1, 4, 7, 11, and 12. The DSFLA tracker is effective in tracking targets with fast movements or when there are blurred images or rotations of the target.

The videos in which all the trackers performed poorly are those with ambient light variation and when the scale of the target has a wide range, as in the case of video 10 (Dog), or moderate occlusion, as in the case of video 15 (Walking2). A common weakness of all the trackers analyzed in this work is related to the variation in ambient lighting. This is probably due to the use of the standardized color histogram to represent the target characteristics. The color histogram is sensitive to any variation in light in the environment, and it can also easily miss the target when the characteristics of the target and background are similar. The illumination of the target in the scene changes substantially and non-proportionally to the frequencies of the histogram since the change in pixel intensity is not linear.

A possible strategy to overcome this problem is to include a target characteristic based on the shape of the object.

When the target and the background have similar characteristics, bounding boxes of different sizes can contain a similar proportion of pixels of the same intensity, and thus, the histograms are similar in appearance. Therefore, the solution space has several local minima whose objective function values are very close. This case can reduce the quality of tracking since candidate targets of different window sizes have a chance of being the estimated target.

The following analyzes check the quality of the RDES model predictions and how much the delimitation of the solution space is useful for tracking. Table 5 shows the RMSE values (in number of pixels) of the forecasts for the x and y coordinates and the Euclidean distance between the forecast $\hat{p}$ and gBest.

Using the data in Table 5 and Equation (31), the 95% confidence intervals for variable RMSEs of x and y are [11.57;49.44] and [11.19;35.87], respectively. Therefore, the prediction error does not exceed 50 pixels of RMSE, that is, the forecasts are reasonably homogeneous and slightly skewed.

Similarly, the 95% confidence interval for the Euclidian distance between the predicted target and gBest is [2.96; 14.48]. Thus, we can conclude that the distance between the predicted and estimated target does not exceed 15 pixels. Considering the largest diagonal of the video image, this value varies from 400 to 800 pixels of the videos observed. Therefore, the biggest forecast error made does not exceed 3.8%, that is, we can conclude that the predictions of the RDES model is quite accurate.

The same experimental design used to calibrate the parameters of the trackers was used to investigate whether video target tracking benefits from restricting the solution space by the proposed region.

Table 6 shows the global average and median of the four videos for the variables’ AUC of success rates per Pascal metric and processing time per frame. Two versions of the DSFLA tracker were assessed: version 1 delimits the solution space, as proposed in this work, and version 2 does not delimit the solution space.

Table 6 shows that the median AUC for version 2 is about 87% of that for version 1 and that the median time to process a frame for version 2 is about 14% longer than that for version 1. Therefore, consistent empirical evidence suggests that the use of the restrictions proposed in this work helps to increase the AUC of the success rate per Pascal metric and improves the processing time.

To conclude, future work to improve the tracker performance involving multiple particle populations acting in different regions of the solution space (in the particle selection process) and an adaptive scheme for quantifying the number of particles to be used in the transfer of knowledge based on the similarity of the frames.

Other representations of the target will also be tested to improve the target recognition ability in environments with varying lighting including the target appearance model with HOG characteristic [35], for instance.

## Figures and Tables

**Figure 1 sensors-21-01903-f001:**
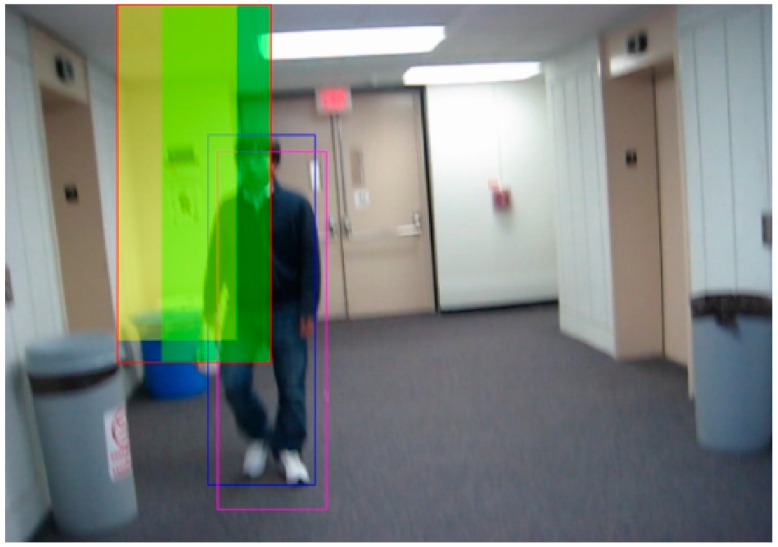
An example of delimiting the solution space obtained from the tenth frame of the video BlurBody (this video was selected from the public Hanyang visual tracker benchmark [44]).

**Figure 2 sensors-21-01903-f002:**
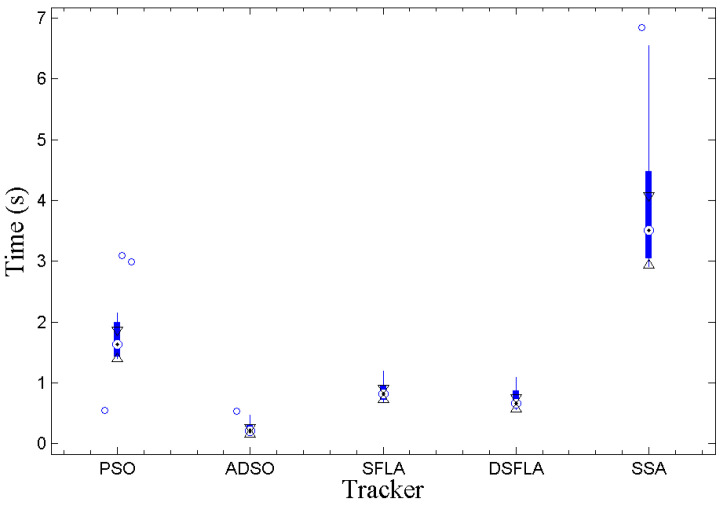
The boxplot of the variable processing time per frame for all trackers.

**Figure 3 sensors-21-01903-f003:**
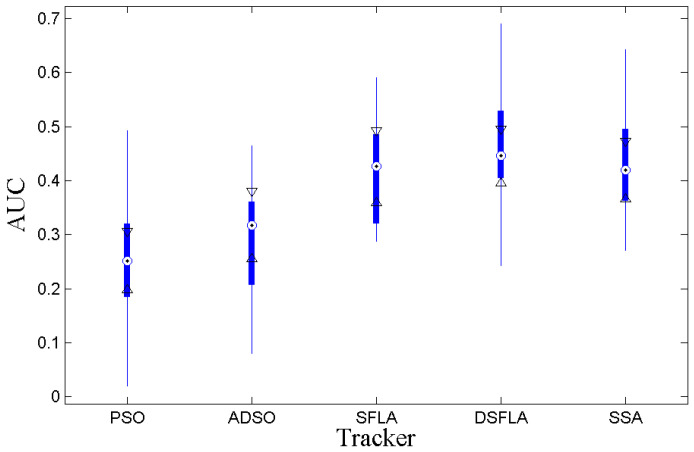
The boxplot of the AUC variable for all trackers.

**Figure 4 sensors-21-01903-f004:**
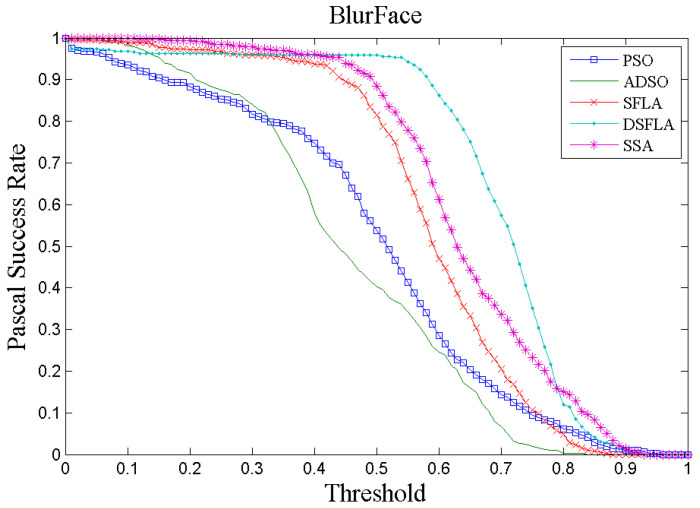
The success rate curve per Pascal metric of video 4 (BlurFace) for all trackers.

**Figure 5 sensors-21-01903-f005:**
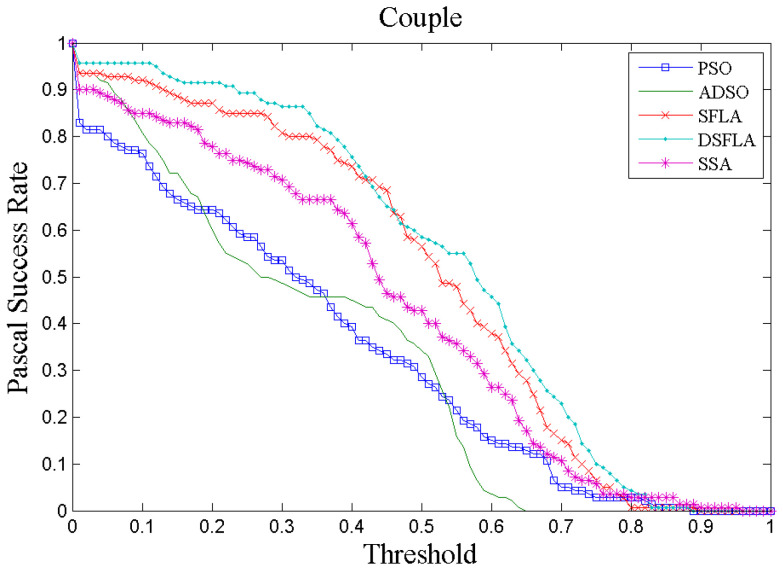
The success rate curve per Pascal metric of video 7 (Couple) for all trackers.

**Table 1 sensors-21-01903-t001:** The features of the trackers.

Trackers	Features	Weaknesses	Strengths
PSO	OP: stationary; SS: continuous; DSS: not; KT: not; PS: not.	unstable results and ambient light variation.	fast processing.
ADSO	OP: dynamic; SS: discrete; DSS: yes. (adaptive); KT: yes; PS: not.	unstable results and ambient light variation.	fast processing.
SSA	OP: stationary; SS: continuous; DSS: not; KT: not; PS: not.	slow processing and ambient light variation.	stable results and fast target movement.
SFLA	OP: Stationary; SS: discrete; DSS: not; KT: yes; PS: not.	ambient light variation.	fast processing, stable results, fast target movement and target rotations.
DSFLA	OP: dynamic; SS: discrete; DSS: yes. (adaptive); KT: yes; PS: yes (adaptive).	ambient light variation.	fast processing, stable results, fast target movement and target rotations.

Note: OP means “optimization problems”; SS means “solution space”; DSS means “delimiting solution space”; KT means “knowledge transfer”; and PS means “particle selection”.

**Table 2 sensors-21-01903-t002:** The features of the videos.

Video	Name	Size	Resolution	Challenges
1	BlurBody	334	(640,480)	IPR, FM, MB, SV, DEF
2	BlurCar2	585	(640,480)	FM, MB, SV
3	BlurCar4	380	(640,480)	FM, MB
4	BlurFace	493	(640,480)	IPR, FM, MB
5	BlurOwl	631	(640,480)	FM, MB, SV
6	Boy	602	(640,480)	IPR, OPR, FM, MB, SV
7	Couple	140	(320,240)	OPR, FM, SV, DEF, BC
8	David2	537	(320,240)	IPR, OPR
9	Deer	71	(704,400)	IPR, FM, MB, LR, BC
10	Dog	127	(352,240)	OPR, SV, DEF
11	Dog1	1350	(320,240)	IPR, OPR, SV
12	Jumping	313	(352,288)	FM, MB
13	MountainBike	228	(640,360)	IPR, OPR, BC
14	Twinnings	471	(320,240)	OPR, SV
15	Walking2	500	(384,288)	LR, SV, OCC
Total	-	6762		

**Table 3 sensors-21-01903-t003:** Processing time per frame (s) (the best results are in bold).

Video	PSO	ADSO	SFLA	DSFLA	SSA
1	3.0914	**0.4776**	1.1673	1.0911	6.8387
2	2.1559	**0.5382**	0.9801	0.9436	4.7558
3	2.9967	**0.2774**	1.1992	1.0951	6.5406
4	2.0773	**0.4318**	0.9154	0.8588	4.6733
5	1.7022	**0.2508**	0.8447	0.7370	3.6917
6	1.4303	**0.2082**	0.8014	0.6653	3.0493
7	1.4158	**0.1959**	0.7841	0.6641	3.0406
8	0.5485	**0.1933**	0.6779	0.5570	2.9147
9	1.7645	**0.3321**	0.9555	0.8803	3.8661
10	1.4918	**0.1947**	0.6860	0.6147	3.2001
11	1.4474	**0.2138**	0.7166	0.6182	3.0866
12	1.4036	**0.1939**	0.7864	0.6572	2.9656
13	1.6324	**0.2263**	0.9009	0.7829	3.5079
14	1.6444	**0.2147**	0.6960	0.6470	3.5379
15	1.6005	**0.1991**	0.8188	0.6511	3.4203
Mean	1.7602	**0.2765**	0.8620	0.7642	3.9393
Median	1.6324	**0.2147**	0.8188	0.6653	3.5079
cv (%)	63.30	**11.48**	16.15	17.25	12.52

**Table 4 sensors-21-01903-t004:** AUC of the success rate per Pascal metric (the best results are in bold).

Video	PSO	ADSO	SFLA	DSFLA	SSA
1	0.4698	0.3654	0.5746	**0.5851**	0.5795
2	0.2518	0.2025	0.3234	**0.3633**	0.3632
3	0.2683	0.0803	0.4033	**0.4091**	0.3791
4	0.4927	0.4558	0.5906	**0.6896**	0.6428
5	0.2347	0.2257	**0.4489**	0.4114	0.4191
6	0.1139	0.3176	0.2882	0.4271	**0.4985**
7	0.3254	0.3198	0.4923	**0.5264**	0.4226
8	0.0206	0.1582	0.2949	0.2436	**0.3244**
9	0.4213	0.4084	0.4261	**0.4874**	0.3623
10	0.1838	0.3494	0.3215	**0.4047**	0.3337
11	0.2435	0.4649	0.4877	**0.5419**	0.4493
12	0.3022	0.1543	0.4668	**0.5300**	0.4838
13	0.2572	0.3229	0.4794	0.4457	**0.4986**
14	0.1905	0.3063	0.4011	**0.4824**	0.3964
15	0.1384	0.2711	0.2973	**0.3423**	0.2708
Mean	0.2609	0.2935	0.4284	**0.4593**	0.4285
Median	0.2518	0.3176	0.4261	**0.4457**	0.4191
cv (%)	49.69	38.20	23.82	**23.63**	23.22

**Table 5 sensors-21-01903-t005:** The RMSE values (in pixels) of the forecasts for the x and y coordinates and the Euclidean distance between the forecast and *gBest*.

Video	*x*	*y*	Distance
1	48.5473	31.7972	17.0507
2	78.8097	33.0285	15.3551
3	99.6405	41.2005	22.3768
4	43.2171	24.7496	20.4736
5	58.8349	81.6870	17.2760
6	30.4878	23.5310	9.6919
7	25.6458	15.6825	6.2391
8	7.2707	7.8175	1.2050
9	67.5223	43.8717	17.0988
10	11.1590	4.8302	2.7205
11	12.1733	9.6148	3.3553
12	41.3441	39.5144	8.7537
13	19.8933	12.0985	4.7146
14	15.1080	5.3767	2.8219
15	3.7789	9.0745	2.4323
Mean	37.5621	25.7911	10.1044
Median	30.4878	23.5310	8.7537
cv (%)	76.24	80.82	73.47

**Table 6 sensors-21-01903-t006:** AUC of the success rates per Pascal metric and processing time for DSFLA tracker with and without delimitation of the solution space (versions 1 and 2, respectively) (the best results are in bold).

	Version 1	Version 2
Mean AUC	**0.3579**	0.3303
Median AUC	**0.3932**	0.3408
Mean time (s)	**0.7193**	0.8092
Median time (s)	**0.6719**	0.7649

## Data Availability

The data was obtained from the public Hanyang visual tracker benchmark [44], an open access CVPR 2013 paper version. site: http://cvlab.hanyang.ac.kr/tracker_benchmark/, accessed on 26 November 2020.

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
