# Peer review of "A New Approach to Enhanced Swarm Intelligence Applied to Video Target Tracking"

_sensors, 2021, doi:10.3390/s21051903_

Round 1

Reviewer 1 Report

Overall Comments,
This article compares the pros and cons of various optimal methods for Video Target Tracking. In many parts of the literature review, it is rather crudely written and not really discussed in depth.
For example, the things mentioned earlier in Target Tracking are found in a bunch of classic books in this area, and there is no mention of the latest technical discussions later on.
From paragraphs 2.1 to 2.3 are overly long and should be more concise and clear.
In the paragraph of "3 Target Tracking", the equation 9 approach is mentioned, but there is nothing new in this section. This is also the traditional way to measure the quality of the predictions.
In the paragraph of "The Proposed Method", the article is quite messy and the method does not seem to be innovative, it feels like just adding some conditionals.
The pseudocode also corroborates the above points. Finally, in the experimental comparisons, Target Tracking was not compared using the more objective standard methods used in many of today's studies.
For example,
https://www.mdpi.com/1424-8220/20/11/3334/htm
https://www.mdpi.com/1424-8220/20/3/929/htm
All of these articles have a more objective approach or presentation. Finally, the Conclusions are written in a rather scattered way.

Author Response

Dear Reviewer,

Greetings.

Please see the attached answer.

Thanks!

Reviewer 2 Report

The authors have proposed a new approach to enhanced swarm intelligence applied to video target tracking. Overall paper is well written but it requires some major revisions before it can be considered for publication.

1) Mention the improvements in tracking time and quality in the abstract. 

2) Clearly mention the novelties of the work in the form of bullets in the introduction section.

3) After the previous work includes a detailed comparison table that should include descriptions, strengths, and weaknesses of the proposed method as compared to the previous methods.

4) Font style of the main heading in section 4 is not consistent with the other headings at the same level.

5) Include more information about the database used. Also, check your method with other publically available datasets.

6) Conclusion is not written appropriately. Please re-write it.

7) Extensive English revisions are also required. Please get it revised from the professional English editing service. 

Author Response

(The authors gave the same response as above.)

Reviewer 3 Report

The current manuscript describes an improvement on the original SFLA aiming at use for video target tracking.

Although the novelty of the method is not very significant, it is very well tailored to the specific application.

I believe the authors should elaborate more on certain aspects of the method, as well as the background:

1) Please fix equation 8. It is not clear how the jump of the worst frog is calculated in each case (first two cases have the same condition).

2) Please explain equation 19 better. What is p-hat in this equation? What means to take the first element of it? Should'nt it have the same dimensionality as gBest?

3) How do you conclude that videos 2, 9, 14 and 15 presented the most difficult challenges for all trackers, based on Table 3?

Author Response

(The authors gave the same response as above.)

Round 2

Reviewer 1 Report

This article compares the pros and cons of various optimal methods for Video Target Tracking.

  1. Authors should describe the proposed model explicitly.
  2. Some mathematical notations and Lemma presentations are not rigorous enough to correctly understand the contents of the paper. The authors are requested to recheck all the definition of variables and further clarify these equations. (equation 1 to 9, 16 to 18, 26 to 29)
  3. In the experimental comparisons, Target Tracking was not compared using the more objective standard methods used in many of today's studies. For example, https://www.mdpi.com/1424-8220/20/11/3334/htm; https://www.mdpi.com/1424-8220/20/3/929/htm; All of these articles have a more objective approach or presentation.

Reviewer 2 Report

Dear Editor,

Most of my comments are addressed, paper can be accepted in the present form. 

Best Regards, 

Reviewer 3 Report

Most comments have been addressed by the authors. One thing that is still not clear is what the notation p-hat_{t-1}(1) means in Equation (23).

Round 3

Reviewer 1 Report

It's okay.